# Illusory finger stretching and somatosensory responses in participants with chronic hand-based pain

**Kirralise J. Hansford**[1]*, **Daniel H. Baker**[1], **Kirsten J. McKenzie**[2], **Catherine E. J. Preston**[1]

**1** Department of Psychology, Faculty of Sciences, University of York, York, United Kingdom, **2** School of Psychology Sport Science & Wellbeing, College of Health and Science, University of Lincoln, Lincoln, United Kingdom

* kirralisehansford@gmail.com

## Abstract

Current pharmaceutical interventions for chronic pain are reported to be minimally effective, leading researchers to investigate non-pharmaceutical avenues for chronic pain treatment. One such avenue is resizing illusions delivered using augmented reality. These illusions resize the affected body part through stretching or shrinking manipulations and have been shown to give analgesic effects; however, the neural underpinnings of these illusions remain undefined. Steady-state evoked potentials (SSEPs) have been studied within populations without chronic pain undergoing hand-based resizing illusions, finding no convincing differences in SSEP amplitudes during illusory stretching. Here, we present comparable findings from a sample with chronic pain, who are thought to have blurred cortical representations of painful body parts, but again find no clear differences in SSEP amplitude during illusory stretching. However, no significant decreases in pain ratings were found following illusory resizing, and changes in SSEP amplitudes are thought to possibly reflect experiences of illusory analgesia. Despite a lack of illusory analgesia across the sample, several participants experienced clinically meaningful levels of pain reduction following illusory resizing, highlighting the potential of resizing illusions as an analgesia treatment avenue. Subjective illusory experience data showed significantly greater experiences of the illusion in the multisensory (visuotactile) condition compared to non-illusion conditions and a unimodal visual condition, replicating findings from participants without chronic hand-based pain. Exploratory analyses using subjective disownership data show that the multisensory condition did not elicit significant disownership experiences, demonstrating that the pain reductions seen in the multisensory condition do not arise from disownership of the limb, but more likely as a direct result of the illusory resizing manipulations.

## 1. Introduction

Chronic pain is classified as pain that lasts or reoccurs for more than 3 months [1, 2] and is a leading cause of disability globally [3]. Current pharmaceutical interventions for chronic pain

**Data Availability Statement:** All data for this project can be found at the following OSF page: https://osf.io/dzmf9/. A script which can be used to

computationally reproduce the entire manuscript, conduct all analyses, and produce all figures can be found at the following GitHub repository: https://github.com/KJHansford/SSEP_illusory_resizing_cp.

**Funding:** A Pain Relief Foundation (https://painrelieffoundation.org.uk/) John Miles PhD Studentship grant funded the work for this manuscript. The grant was awarded to C.E.J.P. The funders did not play any role in study design, data collection or analysis, or the preparation of the manuscript.

**Competing interests:** The authors have declared that no competing interests exist.

conditions are minimally effective, with treatments having ill-defined long-term effects [4], and often being no more effective than placebo at reducing pain or improving functionality [5, 6]. Many drugs prescribed for pain result in around 60% of patients reporting no pain improvement or adverse effects [7, 8]. Surgical interventions to reduce chronic pain can result in up to 34% of patients reporting unfavourable pain outcomes [9]. Due to current treatments being largely ineffective, there is a clear need to find a non-pharmaceutical and non-surgical option for chronic pain treatment.

Individuals who live with chronic pain could have a cortical misrepresentation of their body and its incoming somatosensory signals, including pain [10]. Along with this, it is possible that individuals with chronic pain might experience perceptual size dysfunctions of their affected limbs, which could underpin their persistent pain [10]. There is often reported a lack of concordance between radiographic (physical damage) and symptomatic pain [11, 12]. This highlights the likelihood of a cortical misrepresentation driving pain rather than structural damage, explaining why surgical interventions to treat structural elements of chronic pain could be ineffective. Theories underlying cortical misrepresentations are the predictive coding account [13] and the central sensitisation theory [14, 15]. Predictive coding posits that any mismatch between predicted and actual sensory inputs, such as the difference between peripheral signals and symptomatic pain, generates prediction errors. A lack of updating of top-down expectations in individuals with chronic pain, could lead to constant mismatches between symptomatic and radiographic sensory inputs. Central sensitisation theory, however, refers to the central nervous system changing, distorting, or amplifying pain in a way that no longer reflects the peripheral input from the body, leading to pain becoming an illusory perception [16]. Central sensitisation and predictive coding theories are not positioned in opposition to each other, but rather both contribute to the overall understanding of potential causes of chronic pain conditions. Both theories support the suitability of illusion therapies for the amelioration of chronic pain, as bodily illusions can induce perceptual modulations of the painful body part, altering the patient's perception of their body and the pain related to it.

Illusory resizing is a bodily illusion which changes the way a body part is perceived, exploiting principles of multisensory integration to elicit modulations in the perceived size and shape of the body part [17–19]. Multisensory resizing illusions typically involve both tactile and visual inputs and can be delivered via an augmented reality system. Augmented reality can present real-time video capture of a hand, from the same position and perspective as if the hand were being viewed directly [17], allowing the experimenter to deliver tactile manipulations, such as gently pulling the hand / fingers, whilst the participant views their hand / fingers stretching in the augmented image. Newport, Pearce and Preston [20] found strong embodiment using multisensory visuotactile illusions, and our recent work [21] found that multisensory illusions elicited significantly greater illusory experience compared to non-illusion conditions. Regarding unimodal visual illusions, which consist of visual input of the finger stretching but without any tactile input, mixed results have been found with inconsistencies reported in illusory experience. Some participants show quite high illusory experiences during unimodal visual presentations, whilst others report no experience of illusory stretching [21, 22]. Previous research has found a reduction in hand and knee pain in osteoarthritis (OA) patients using augmented reality to deliver multisensory resizing illusions [17–19], therefore both multisensory and unimodal visual resizing illusions are delivered in the present study to assess if illusory experience is required for illusory analgesia.

There are two main theories underlying analgesic resizing illusions. Firstly, the somatosensory blurring hypothesis posits that the cortical representation of a painful body part is blurred, and viewing the body part sharpens this representation. This is supported through findings from participants without chronic pain, where visual analgesia has been found following

experimentally induced pain [23]. The second theory stems from research by Gilpin et al. [24], finding that participants with arthritis make smaller hand judgements compared to those without the condition, suggesting a reduced cortical representation of their hands. Pain reductions have been found for participants with arthritis when using stretching resizing illusions [17], therefore, Gilpin et al. [24] posit that increasing cortical representation of the hands through magnifying (stretching) could reduce pain. Both theories predict that cortical misrepresentations occur at the somatosensory cortex, with both theories predicting different neural changes regarding the experience of pain. Specifically, the somatosensory blurring hypothesis predicts a larger, more diffuse representation of the painful body part that would be reduced (sharpened) during resizing illusions, whereas the magnification theory predicts a shrunken representation of the painful body part that would be enlarged following illusory stretching.

Somatosensory cortex modulation has been investigated using steady-state evoked potentials (SSEPs), where low-level somatosensory responses can be induced directly using vibrations of a known frequency applied to a body part. These generate a frequency-locked steady-state evoked potential detectable at the scalp using EEG [25, 26], and are an index of the cortical response to a stimulus, therefore can give an index of cortical response changes during illusory resizing. Our previous work [22] despite finding slight steady-state response decreases when participants without chronic pain underwent resizing illusions, gave no convincing evidence of somatosensory sharpening in participants without chronic pain. Since people with chronic pain are thought to have cortical misrepresentations of their affected body parts, it is plausible that using the same paradigm as in our previous work, we might see greater differences in somatosensory response to illusory stretching in a population with chronic hand-based pain. SSEP responses can therefore be used to directly compare the somatosensory blurring hypothesis [23] and the magnifying hypothesis [24], as an increased SSEP response following illusory resizing could indicate support for the magnification hypothesis, suggesting increased cortical representation of the painful body part, whereas a smaller SSEP response after illusory resizing could support the somatosensory blurring hypothesis, suggesting the cortical representation of the body part has become sharpened.

Using different sensory manipulations of finger resizing illusions, in addition to using an electromagnetic solenoid stimulator to elicit SSEPs, this study aimed to investigate subjective illusory experience and neural responses to resizing illusions in participants with chronic hand-based pain. To test this, different resizing illusions consisting of multisensory (visuotactile) stretching (MS), unimodal-visual stretching (UV), a non-illusion control condition without tactile input (NI), and a non-illusion control condition with tactile input (NIT) were used. Previous research has suggested that tactile input alone can reduce pain ratings [27, 28], therefore this second control condition was used to demonstrate if the illusion itself delivered analgesia rather than the tactile or combined sensory inputs. This study had three categories of hypothesis, the first relating to illusory experience (1), the second related to SSEP responses (2a-c), and the third to pain reduction (3a-c). The first hypothesis, acting as a positive control (1), was that there would be a greater illusory experience, measured via a subjective illusory experience questionnaire, in the MS condition compared to the NI and NIT conditions. The main experimental hypothesis was that (2) there would be a significant difference in SSEP response when comparing (2a) MS illusory resizing to the NI condition, when comparing (2b) UV illusory resizing to the NI condition, but no difference when comparing (2c) the NIT condition to the NI condition. The final hypothesis was that (3) there would be a reduction in pain, measured via a 21-point numeric rating scale, comparing before and after scores for (3a) MS and (3b) UV conditions, whilst we expected (3c) no reduction of pain following the NI condition, nor (3d) a reduction of pain following the NIT condition.

## 2. Methods

### 2.1 Preregistration

This study was preregistered at the following OSF page: https://osf.io/9anjc. Deviations from the preregistration are as follows:

- To prevent bias towards participants with a diagnosis of a chronic pain condition, participants without diagnoses were recruited as long as they self-reported to be experiencing ongoing or reoccurring pain for more than 3 months.

- Due to scarcity of participants eligible to take part in this project within the timeframe, additional participants were not recruited to replace any lost due to incomplete or noisy data.

### 2.2 Sample size

Based on power analyses in section 2.5, a sample size of 30 participants was aimed for to adhere to the higher end of sample size estimates (Hypothesis 2 (2.5.2)). However, due to scarcity of participants experiencing pain in either their right index or middle fingers (digits needed for the delivery of vibrotactile and illusory manipulations, see section 2.2 Sample inclusion / exclusion criteria), a final sample size of 21 participants (mean age = 48.8 years; age range = 19–73 years; sex = 22.7% male, 77.3% female; ethnicity = 95% white; chronic pain = 5 primary pain, 10 secondary pain, 2 mixed, 4 no diagnosis) were tested after an 8-month recruitment period (starting 01/12/2023 and ending 31/07/2024).

### 2.3 Participants

Ethical approval was gained from the Department of Psychology, University of York (ethics application code 950), in line with the Declaration of Helsinki. Informed written consent from each participant was gained prior to the start of any experimental set up, and participants were instructed that they could withdraw their participation at any time during or after completion of the experiment.

Sample inclusion / exclusion criteria:

Inclusion and exclusion criteria were determined using self-report responses relating to each item listed below:

- Inclusion Criteria: Right-handed, over 18 years of age, must have ongoing or reoccurring pain in their right index or middle fingers (or their associated joints) for more than 3 months, hand-based pain present on day of testing. No formal diagnosis of a chronic pain condition was needed*, as this has been found to be a barrier for participants taking part in non-pharmaceutical chronic pain research studies, especially for individuals from ethnic minorities [29].

\* All participants were asked whilst giving consent to take part if they had any chronic pain condition diagnoses. Diagnoses were then categorised into either chronic primary pain conditions, chronic secondary pain conditions, those with a mixture of primary and secondary conditions, and those without a diagnosis for the purpose of exploratory chronic pain condition analyses.

- Exclusion Criteria: Prior knowledge or expectations about the research, a history of developmental, neurological or psychiatric disorders, history of drug or alcohol abuse, history of sleep disorders, history of epilepsy, visual abnormalities resulting in complete visual occlusion, being under 18 years of age, diagnosed with Complex Regional Pain Syndrome. No

restrictions applied regarding any medication the participant might be taking. (Participants with a diagnosis of Complex Regional Pain Syndrome were not recruited to take part in the study due to research showing increasing pain after stretching illusions [30]).

Raw data exclusion criteria:

- Less than 100% of the experiment completed by a participant, more than 50% of electrodes for a single participant requiring removal from EEG data, or if both electrodes F1 andFC1 (electrodes of interest) required removal. More information about data removal can be found in section 2.4.1 Preprocessing Steps.

## 2.4 Experimental procedure

All participants filled out a demographic survey, asking their age, sex, ethnicity, and any chronic pain condition diagnosis, and were asked to complete the revised Waterloo Handedness Questionnaire (WHQr) [31]. The WHQr self-reported handedness questionnaire consisted of 36 questions. The questions were answered on a 5-level Likert scale to determine the degree of preferred hand use, with left always being -2, left usually being -1, equal use being 0, right usually being +1 and right always being +2. The sum of the total WHQr score was then used to categorise a respondent as left-handed (score of -24 or less), mixed handed (score of -23 to +23), or right-handed (score of +24 or higher). Only participants who were categorised as right-handed continued participation, with the exception of one participant who scored a result of mixed handed due to changes in hand use as a result of their pain. Participants were asked their pain score on the day of testing for their digit in the most pain using a 21-point numeric rating scale (NRS) (0 = no pain at all; 20 = most severe pain imaginable). This 21-point scale has equivalent reliability to a more frequently used 11-point scale [32] and was chosen to aid comparability with previous studies which have used the 21-point NRS [17, 18]. Additionally, since the scale is different to a typical rating scale of 1–10, participants would be more likely to think about the answer they give, rather than giving a number they always use when asked to rate their pain on a scale of 1–10. Participants were only tested if their pain on the day was above 0 on the 21-point scale.

Participants were set up with an appropriately sized 64-channel EEG cap with electrodes arranged according to the 10/20 system. The experimenter used conductive gel to make a conductive bridge between the electrodes and the scalp to attempt to obtain impedance levels of $<10k\Omega$ per electrode. The whole head average was used as a reference.

Participants were seated behind the augmented reality system (Fig 1) and instructed to place their hand onto the black felt fabric within the augmented reality system. Within the self-built system there is a 1920 x 1080 Spedal Webcam Wide Angle Camera situated in the middle of the central area, away from the participant's view. 26cms above the felt base of this central area, there is a mirror, which is placed 26cms below a 1920 x 1200 resolution screen, with a width of 52cms. This screen is 54cms from the base of the system, and the base of the system is 82cms from the ground. Participants were asked which digit was in the most pain and were asked to place this digit outstretched onto the felt. If multiple digits were equally painful, the digit that the participant chose as their preference was used. There were two white dots for each hand on the felt and participants were instructed to place their right hand between the two right dots. Participants were instructed to view their hand's image in the mirror (the real hand was hidden from view) throughout the experiment. The camera placed underneath the mirror on the felt base was used to deliver a live feed video of the participants hands to the computer screen at the top of the augmented reality system, which showed in the mirror reflection to the participants. There is a delay of 170ms in the video processing pipeline from the camera image to the augmented video image.

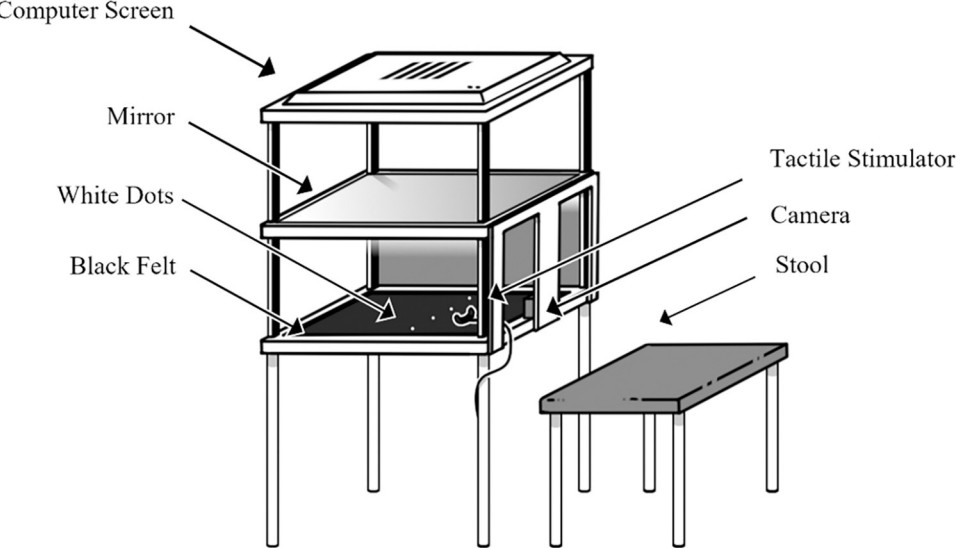

**Fig 1. Schematic of augmented reality system with tactile stimulator.**

Participants underwent 4 conditions: multisensory stretching (MS), unimodal-visual stretching (UV), a non-illusion control condition without tactile input (NI), and a non-illusion control condition with tactile input (NIT). There was vibrotactile stimulation to the finger in all conditions, but only tactile input of the researcher touching the participants hand in the conditions where this is mentioned. Tactile input was given in the NIT condition due to previous research suggesting that tactile input alone can reduce pain ratings [27, 28]. Each trial lasted 2.4 seconds for the manipulation phase, where the finger was stretched by 60 pixels (2.1 centimetres) in UV and MS conditions, followed by a further 2.4 second habituation phase in which participants could view and move their augmented finger before the screen went dark, indicating that the next trial can start. MS conditions consisted of the researcher touching and pulling the participant's finger as the participant viewed their finger stretching in a congruent manner. UV conditions consisted of the participants viewing their finger stretch without any experimenter manipulation. The NI condition provided no visual or touching tactile manipulations to the finger. The NIT condition involved no visual input of the finger stretching, instead the image of their finger was visible but unchanged whilst tactile input of the experimenter's hand touching the participant's finger, but without pulling, was applied. Visualisation of all conditions can be seen in Fig 2.

The experimenter was seated opposite the participant, the other side of the augmented reality machine and pulled the digit by holding onto the distal interphalangeal joint and gently pulling the finger whilst the participant kept their hand in place. Conditions were delivered across 4 blocks, with each block consisting of 24 trials of the same experimental condition, totalling 96 trials over all 4 blocks. The ordering of the blocks was randomised for each participant to prevent ordering effects. The experiment was programmed in, and the conditions randomised using MATLAB R2017a and the experimenter was informed of whether to pull the finger or to apply no manipulation via an indicative box displayed on the screen out of the participant's view. If the box was blue, this indicated a need to pull the finger, if it was white it indicated a need to touch the finger. The researcher used a button press to dictate the start of the manipulation, and started pulling the finger, when needed, synchronously within the 2.4 second manipulation phase. This occurred 1 second after the researcher pressed the button to

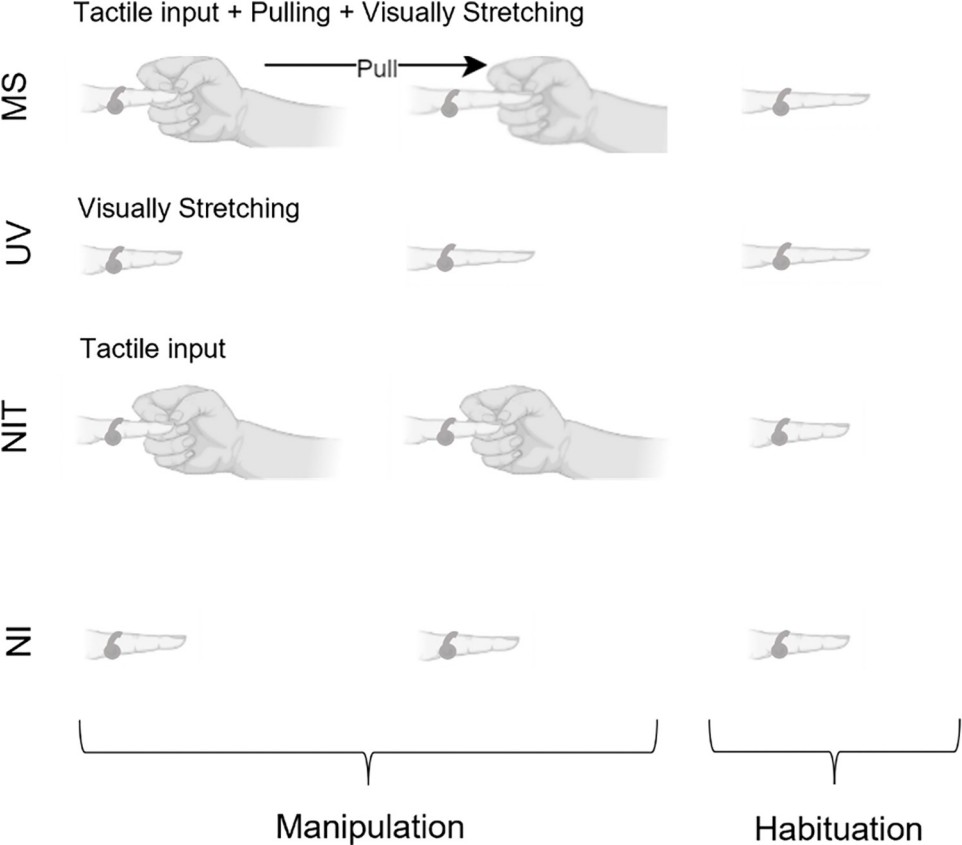

**Fig 2. Infographic of experimental conditions.** MS = Multisensory Stretching, UV = Unimodal Visual Stretching, NIT = Non-Illusion Tactile, NI = Non-Illusion. During the manipulation phase (2.4 seconds) the visual image of the finger is stretched in the MS and UV conditions, and/or the experimenter provides tactile input in the MS and NIT conditions,. The tactile input in the MS condition is accompanied by pulling. During the habituation phase (2.4 seconds) participants are free to move their finger. The arrow denotes the direction of the experimenter's action. The vibrotactile stimulator is depicted on the finger in each phase of the experiment as vibrations are presented throughout.

initiate the trial. Vibrations were delivered to the participant's finger in all conditions using a miniature electromagnetic solenoid stimulator/bone conductor (Dancer Design Tactor; diameter 1.8mm) emitting vibrations produced by sending amplified 26Hz sine wave sound files, with stimulus intensity controlled by an amplifier (Dancer Design TactAmp). 26Hz was the chosen frequency after pilot testing (see S1 Fig). The tactor was driven at 50% of the maximum (i.e. a peak input voltage of 3V) using a 26Hz sine-wave, and delivered a peak force of 0.18N. The electromagnetic solenoid stimulator was attached to the participant's finger that was outstretched, between the knuckle and the first finger joint, using clear medical tape and gave continuous stimulation for the duration of each trial. Participants were asked before each condition block and then again immediately after each condition block to rate their pain on the 21-point NRS, which was a verbal report that the experimenter inputted onto a Samsung Galaxy A6 Tablet, resulting in 4 pre and 4 post block pain reports per participant. Participants were encouraged to take a break between each of the blocks to stretch their hand. All participants were naïve to resizing illusions before taking part in the experiment. EEG was recorded throughout as a continuous recording with trials and conditions denoted by numbered 8-bit digital triggers sent when the researcher pressed a button box to start each trial (USB-TTL Module, Black Box Toolkit Ltd.).

Finally, at the end of each block, the participant was asked to complete a subjective illusory experience questionnaire regarding a condition presented in a given block using the Samsung Galaxy Tab A6 tablet via a questionnaire on Qualtrics (Qualtrics, Provo, UT). The questionnaire consisted of six questions relating to the trial the participant had just experienced. Two statements related to illusory experience (ownership): "It felt like my finger was really stretching" / "It felt like the finger I saw was part of my body", two related to disownership: "It felt like the finger I saw no longer belonged to me" / "It felt like the finger I saw was no longer part of my body", and two were control questions: "It felt as if my finger had disappeared" / "It felt as if I might have had an extra finger" (all questions were directed towards the participants manipulated finger). Control questions were included to create an index for the illusion and disownership questions (more detail can be found in section 2.4.1—Preprocessing steps), whilst disownership questions were included to assess if the potential experience from the illusions resulted from a disownership of the body part, or from subjective embodiment of the body part [33]. A visual analogue scale from 0–100 was used for each statement, with 0 being strongly disagree, 50 being neutral and 100 being strongly agree.

Data collection was terminated after 8 months of recruitment. If a participant needed over 50% of the electrodes removed during preprocessing, or if both electrodes F1 and FC1 needed removal, then their data was not included for SSEP analysis, which was the case for one participant. Due to difficulties recruiting participants with hand-based pain affecting the right index and/or middle digits, no additional participants were recruited to replace lost data.

## 2.5 Analysis pipeline

**2.5.1 Preprocessing steps.** EEG data were first converted using MATLAB and EEGlab from the ANT EEprobe.cnt format to EEGlab.set format. All subsequent analysis was then conducted using the MNE-Python toolbox [34]. A 50Hz notch filter was first applied to the raw EEG data for all electrodes, followed by calculation of the standard error across time for each electrode for each participant [35]. Across the standard errors for all participants, the 5% of electrodes which showed the largest standard errors were used to create a standard error threshold. Any electrode with a standard error above this threshold, or with a value of 0, was removed from analysis. Where a participant had over 50% of their electrodes over the standard error threshold or with a value of 0, or if the electrodes requiring removal contain both electrodes F1 and FC1 (electrodes of interest), then their data was removed. Primary analysis of the remaining EEG data then involved averaging the signal across the electrodes of interest (or using just electrode F1 or FC1 in case of electrode removal), and calculating the Fourier transform for each trial per participant. These amplitudes were then averaged across trials to give overall results for each participant. Statistical comparisons were then performed on the Fourier amplitudes at the stimulation frequency (26Hz), across conditions and participants. No additional filtering or denoising steps were applied to the EEG data, in line with Figueira et al.'s [36] report that only a Fourier transform is typically needed for this type of EEG data.

Regarding questionnaire data, scores for both illusion experience questions were combined to give median scores, along with both disownership questions and both control questions, resulting in 3 median scores per condition per participant. The median control scores were used to create an index of the illusion and disownership scores by subtracting the median control score from the median illusion and median disownership scores, in line with previous research doing similarly [37–39]. The normalised (baseline corrected) data were used for analyses, with a new scale from -100 to +100 with 100 indicating strongly agree, 50 indicating a neutral opinion, and scores below 0 indicating strongly disagree with the statements on the questionnaire. 50 is maintained as a neutral opinion so that the normalised data still adhered to the thresholds that the participants were presented with during the experiment.

8 data points were collected per participant for their pain ratings. Median scores were then calculated across pain data for pre and post scores for all experimental conditions.

## 2.6 Power analyses & analysis plans

**2.6.1 Hypothesis 1 (positive control).** (1 –Positive Control) There will be a greater illusory experience, measured via a subjective illusory experience questionnaire, in the (1a) MS condition compared to the NI condition and in the (1b) MS condition compared to the NIT condition.

The subjective illusory experience questionnaire was used as a positive control for the current study. Previous research has shown significantly greater illusion strength for MS conditions compared to non-illusion conditions [21, 22, 40], which we attempted to replicate. Questionnaire data was analysed using R [41] in line with preregistered analysis plans (https://osf.io/9anjc).

Effect sizes were determined by research from Hansford et al [21] using the subjective illusory experience questionnaire and comparing MS, UV, and incongruent finger-based resizing illusions to control conditions with no illusory resizing, using the same finger stretching illusions and the same equipment (n = 48), which showed an effect size of η 2 = .33 (converted to a Cohen's f = .70). Additional effect size information came from a visual capture study (n = 80) using a subjective embodiment questionnaire and visual and tactile manipulations to a mannequin body [40], showing an effect size of r = .64 (converted to a Cohen's f = .83) when comparing embodiment scores from the questionnaire against control scores. An effect size of f = .70 was used for hypothesis 1 to adhere to the lower end of previous effect sizes.

A priori power analysis using G*Power for the smallest effect size of interest (f = .70) showed that for a repeated measures, within factors one way ANOVA, with an effect size (f) of 0.70, alpha of 0.05, power at 90% and 1 group with four measurements, 6 participants were needed.

**2.6.2 Hypothesis 2.** (2) There will be a significant difference in SSEP response when comparing (2a) MS visuotactile illusory resizing to the NI condition, when comparing (2b) UV illusory resizing to the NI condition, and when comparing (2c) the NIT condition to the NI condition.

As mentioned in the EEG pre-processing steps in section 2.4.1, EEG data analysis involved taking a Fourier transform for each waveform averaged across the electrodes of interest, to obtain the amplitude for each trial at the vibration frequency (26Hz). These amplitudes were then averaged across trials to give overall results for each participant, before following preregistered analysis plans (https://osf.io/9anjc). Based on the pilot data in S1 Fig, we expected to see activation most pronounced over mid-frontal distributions, covering F1 and FC1 electrodes and therefore these electrodes were selected as the electrodes of interest.

Despite our previous work using SSEPs to assess somatosensory response changes during illusory finger stretching, this was the first study to investigate illusory finger stretching using SSEPs in a chronic pain sample, so appropriate effect size estimates were not available. We therefore conducted power calculations based on a smallest effect size of interest, in line with the recommendation of Lakens [42]. Here, we chose an effect size of d = 0.5 (a medium effect, [43]), since this is the smallest effect size we were interested in detecting, which converted to a Cohen's f of 0.25 for power analyses.

A priori power analysis using G*Power showed that for a repeated measures, within factors one way ANOVA, with an effect size (f) of 0.25, alpha of 0.05, power at 90%, and 1 group with four measurements, a total sample size of 30 participants was needed.

**2.6.3 Hypothesis 3.** (3) We expect to find a subjective reduction in pain, measured via a 21-point numeric rating scale, comparing before and after scores for (3a) MS and (3b) UV

conditions whilst we expect (3c) no reduction of pain following the NI condition, nor (3d) a reduction of pain following the NIT condition.

Pain data were also analysed using R [41] following preregistered analysis plans (https://osf.io/9anjc). Comparisons of the MS and the NIT conditions assessed whether any reduction in pain was due to the illusory manipulations or rather, due to the addition of tactile input.

Effect size was determined using those listed in previous research using the 21-point numeric pain rating scale [18] and from previous pilot data using the same MS resizing illusions for analgesic effect, finding post illusion pain scores to be significantly lower than pre illusion scores (t(10) = 3.32, p = .008, $d$ = 1.0).

A priori power analysis using G*Power showed that for a Wilcoxon signed-rank test (one-sided, matched pairs), with an effect size (dz) of 1, alpha of 0.05, and power at 90%, for a two tailed test with normal parent distribution, 11 participants were needed in total.

## 2.7 Data and code availability

All data for this project can be found at the following OSF page: https://osf.io/dzmf9/. A script which can be used to computationally reproduce the entire manuscript, conduct all analyses, and produce all figures can be found at the following GitHub repository: https://github.com/KJHansford/SSEP_illusory_resizing_cp.

## 3. Results

Positive control analyses of the subjective illusion data can be seen in Fig 3. A Friedman test found a significant overall effect of condition with a moderate effect size ($\chi^2$ (3) = 16.44, $p$ = <0.001, Kendall's W = 0.26) and post hoc Wilcoxon tests with Holm corrections found significantly greater combined illusion score in the Multisensory Stretching (MS) condition (Median = 67.5, SD = 30.16) compared to the Non-Illusion (NI; Median = 49, SD = 27.62, $z$ = 13.5, $p.adj$ = 0.003, $r$ = -26.14) and Non Illusion Tactile (NIT; Median = 50, SD = 21.66, $z$ = 29, $p.adj$ = 0.019, $r$ = -26.25) conditions, thereby supporting hypotheses 1, 1a, and 1b and fulfilling the positive control checks. Exploratory analysis also showed a significant difference between the MS and Unimodal Visual (UV) condition (Median = 20.5, SD = 43.02, $z$ = 204, $p.adj$ = 0.001, $r$ = 40.11).

Exploratory analysis of subjective disownership and control data can be seen in S3 and S4 Figs. A significant difference in disownership scores was found between the UV condition (Median = 42, SD = 42.32) compared to the NI (Median = 0, SD = 19.63, $z$ = 16, $p.adj$ = 0.039, $r$ = -39.78), NIT (Median = 0, SD = 15.24, $z$ = 12, $p.adj$ = 0.025, $r$ = -48.25), and MS conditions, (Median = 1.5, SD = 22.34, $z$ = 22, $p.adj$ = 0.042, $r$ = -40). Regarding control data, a significant difference was found between NI (Median = 0, SD = 21.8) and UV (Median = 6, SD = 24.02) control scores ($z$ = 1, $p.adj$ = 0.031, $r$ = -16).

Analyses of SSEP data can be seen in Fig 4B. The left panel (a) confirms the presence of a clear steady-state signal at 26Hz, which was strongest over fronto-central electrodes. A Friedman test found no significant overall effect of condition with a small effect size ($\chi^2$ (3) = 2.4, $p$ = 0.494, Kendall's W = 0.04) opposing Hypothesis 2. Despite the MS condition having numerically the lowest median amplitude (Median = 0.16, SD = 0.29), post hoc Wilcoxon tests with Holm corrections found no significant differences between SSEP amplitude when comparing the NI condition (Median = 0.21, SD = 0.38) to the MS condition ($z$ = 100, $p.adj$ = 1.000, $r$ = -0.01), or the UV condition (Median = 0.2, SD = 0.26, $z$ = 95, $p.adj$ = 1.000, $r$ = -0.01), meaning Hypotheses 2, 2a, and 2b were unsupported. There was no significant difference found when comparing the NI condition to the NIT condition (Median = 0.18, SD = 1.71, $z$ = 99, $p.adj$ = 1.000, $r$ = 0), supporting Hypothesis 2c.

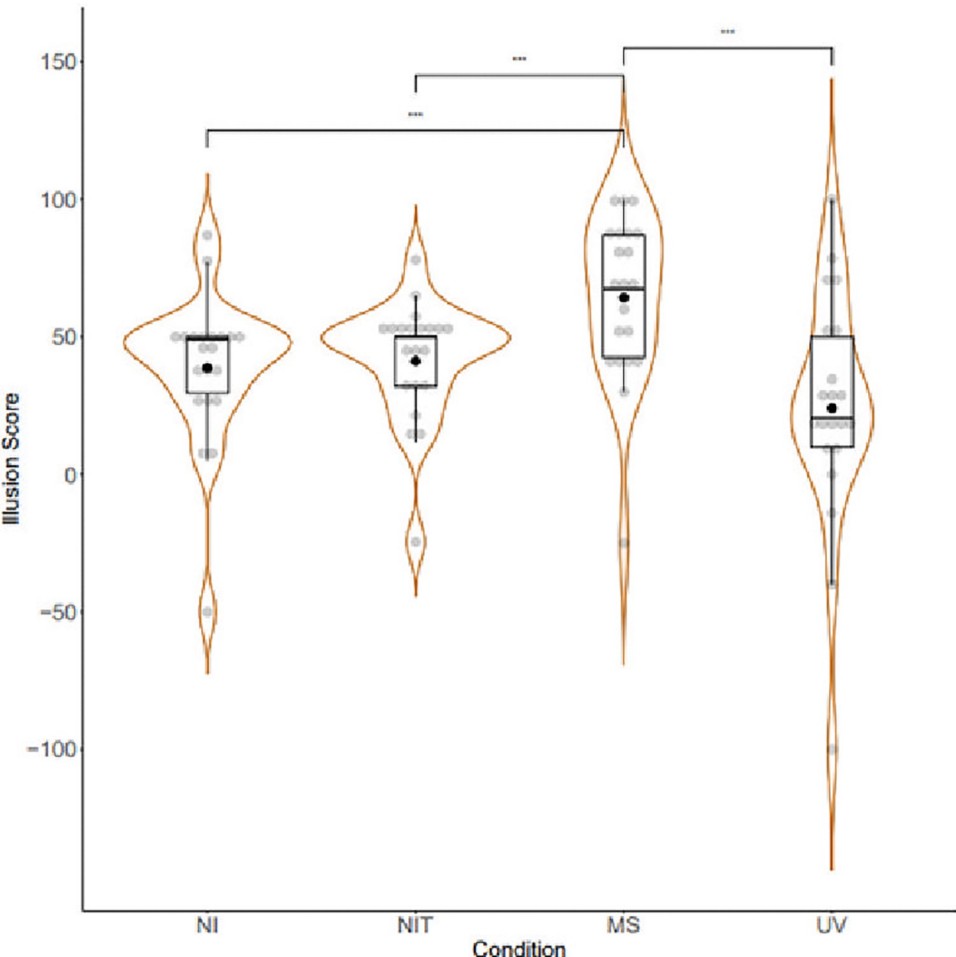

**Fig 3. Combined illusion score index across conditions (NI: Non-Illusion; NIT: Non-Illusion Tactile; MS: Multisensory; UV: Unimodal Visual).** Scores below 50 indicate disagreement with experience of illusion statements, whilst scores above 50 indicate agreement. A continuous visual analogue scale was used in data collection, with agreement and disagreement statements located at each end of the scale. Box plots show means, medians and interquartile ranges of data. Medians are indicated with a horizontal line whilst means are indicated by a black dot. Data points are shown in grey jitter binned along the y-axis, grouped by condition.

Exploratory correlation analyses were conducted to assess the correlation between participant's subjective illusion score and their SSEP amplitude across electrodes of interest (F1 & FC1) for each condition to see if those who experienced a stronger feeling of the illusion had more reduced SSEP amplitudes, results showed no significant correlations. Exploratory correlation analyses and figures can be found in S5–S7 Figs.

Analysis of pain data across conditions can be seen in Fig 5. Wilcoxon tests found a significant increase in pain when comparing NI pre (Median = 4, SD = 3.27) and post (Median = 6, SD = 4.23) pain levels ($z$ = 113, $p.adj$ = 0.021, $r$ = 2). No significant differences were found when comparing NIT pre (Median = 4, SD = 3.25) and post (Median = 5, SD = 4.02) pain levels ($z$ = 101.5, $p.adj$ = 0.243, $r$ = 0.75), in line with our hypotheses. No differences in pain were found when comparing the MS pre (Median = 4, SD = 3.38) and post (Median = 5, SD = 4.61) levels ($z$ = 85.5, $p.adj$ = 1.000, $r$ = 0) nor the UV pre (Median = 5, SD = 2.56) and post (Median = 5, SD = 3.5) levels ($z$ = 105, $p.adj$ = 0.403, $r$ = 0.5) levels, opposing our hypotheses.

Despite finding no significant differences in pain levels for illusory conditions when conducting group level analyses, there were cases of pain reduction for each condition for

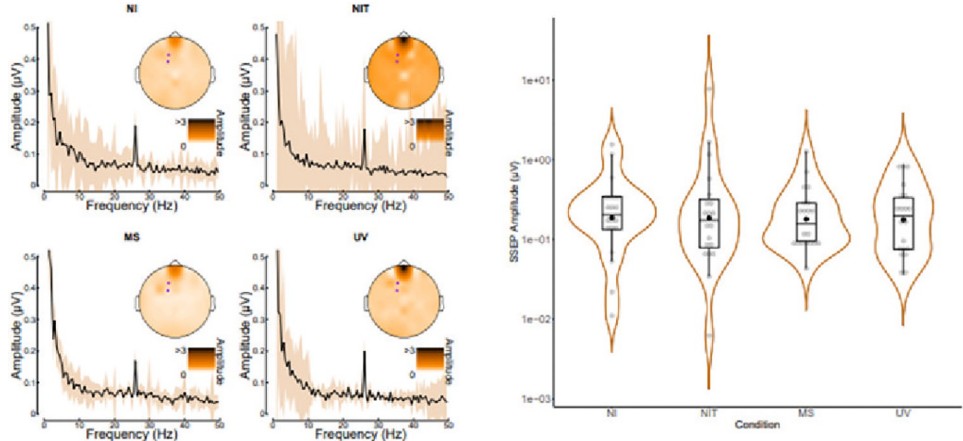

**Fig 4.** (a): SSEP Amplitude spectra across conditions (NI: Non-Illusion; NIT: Non-Illusion Tactile; MS: Multisensory; UV: Unimodal Visual) for electrodes of interest (F1 & FC1). Shading shows ±1 standard error across participants (n = 46). (b): SSEP Amplitudes Across Conditions. Box plots show means, medians and inter-quartile ranges of data. Medians are indicated with a horizontal line whilst means are indicated by a black dot. Data points are shown in grey jitter binned along the y-axis, grouped by condition.

individual participants. The MS condition resulted in 9 participants experiencing a reduction in pain, with 7 participants experiencing a reduction greater than 30%, which is described as a clinically meaningful level of pain reduction [44] and 6 greater than 50% (described as extremely meaningful). The UV condition resulted in 9 participants experiencing a reduction in pain, 4 of which experienced a reduction greater than 30% and 1 a reduction greater than 50%. These conditions saw more participants experience reductions in pain levels compared to the non-illusion conditions which saw only 3 participants experience a reduction in pain following the NI condition, 1 of which experienced a reduction greater than 30% and 1 a reduction greater than 50%, and 6 participants experience a reduction in pain following the NIT condition, with 5 participants experiencing a reduction greater than 30% and 2 participants experiencing a reduction greater than 50%. Fig 6 shows percentage change per participant per condition, showing some participants experiencing a reduction in pain levels along with some experiencing no change in their pain levels (NI: 4, NIT: 3, MS: 3, UV: 2) and others showing increases in pain following each condition (NI: 13, NIT: 11, MS: 9, UV: 9).

To assess if participants experiencing a reduction in pain differed between presentations of chronic primary and secondary pain conditions, data were analysed split by condition type and can be found in the S8–S10 Figs. No significant differences were found when comparing pre and post pain levels across any condition for either chronic primary or secondary pain.

Exploratory correlations were also run between participants pain percentage change and their SSEP amplitude across conditions, to assess if those experiencing a reduction in pain showed a lower SSEP amplitude, however no significant correlations were found. Further exploratory correlations were run across conditions comparing participant's pain percentage change data and their subjective illusion scores, also finding no significant correlations. All exploratory correlation analyses can be seen in S3 Fig.

## 4. Discussion

This study investigated subjective illusory experience and neural response to resizing illusions in participants with chronic hand-based pain to assess whether illusory resizing of the fingers would reduce pain levels and show differences in steady state responses to 26Hz vibrotactile

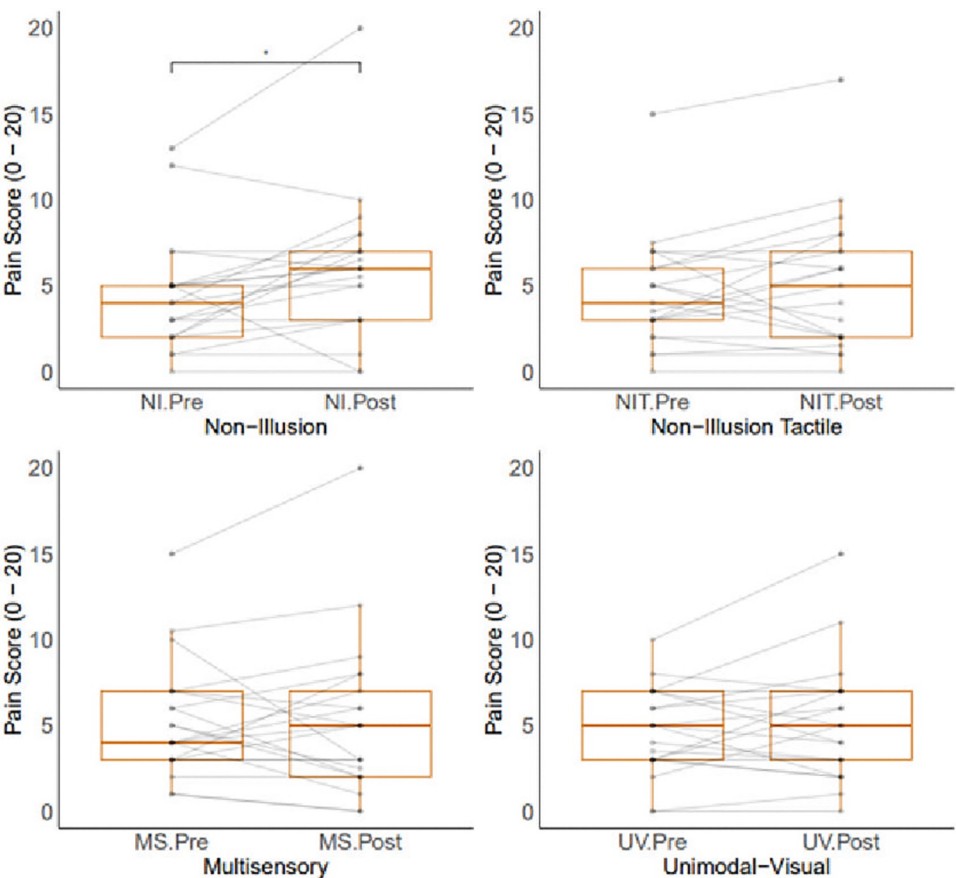

**Fig 5.** (a): Pre and post pain scores across conditions. Box plots show medians and interquartile ranges of data. Paired data points are shown in grey.

stimulation across illusory conditions. Subjective results replicated previous findings in samples without chronic pain of a significantly greater experience of the illusion in the multisensory condition compared to both non-illusion conditions, with the unimodal visual condition showing a wider range of illusory experience responses. SSEP responses to 26Hz vibrotactile stimulation showed no significant differences in response amplitude across conditions opposing our hypothesis, with pain ratings also showing no significant differences when comparing pre and post levels for both illusory conditions, contrasting previous analgesic findings.

Our work delivering resizing illusions to participants without chronic pain [22] showed surprising effects of a significantly greater experience of the illusion in the multisensory condition compared to the unimodal visual condition, despite previous findings of visual capture alone being found to elicit embodiment in both virtual [45] and physical environments [40], and when viewing an illusion of an elongated arm [46] or from visual-only manipulations of the hand [47]. Previously, we found significantly greater levels of disownership during unimodal conditions compared to multisensory conditions and therefore posited that the tactile input of touching the hand / finger is needed to ground one's experience of owning their body part within augmented reality. The present study replicated these findings of significantly heightened disownership levels during the unimodal visual conditions, further supporting the idea that tactile input grounding one's experience could contribute to greater experiences of illusory stretching, regardless of the presence of a chronic pain condition.

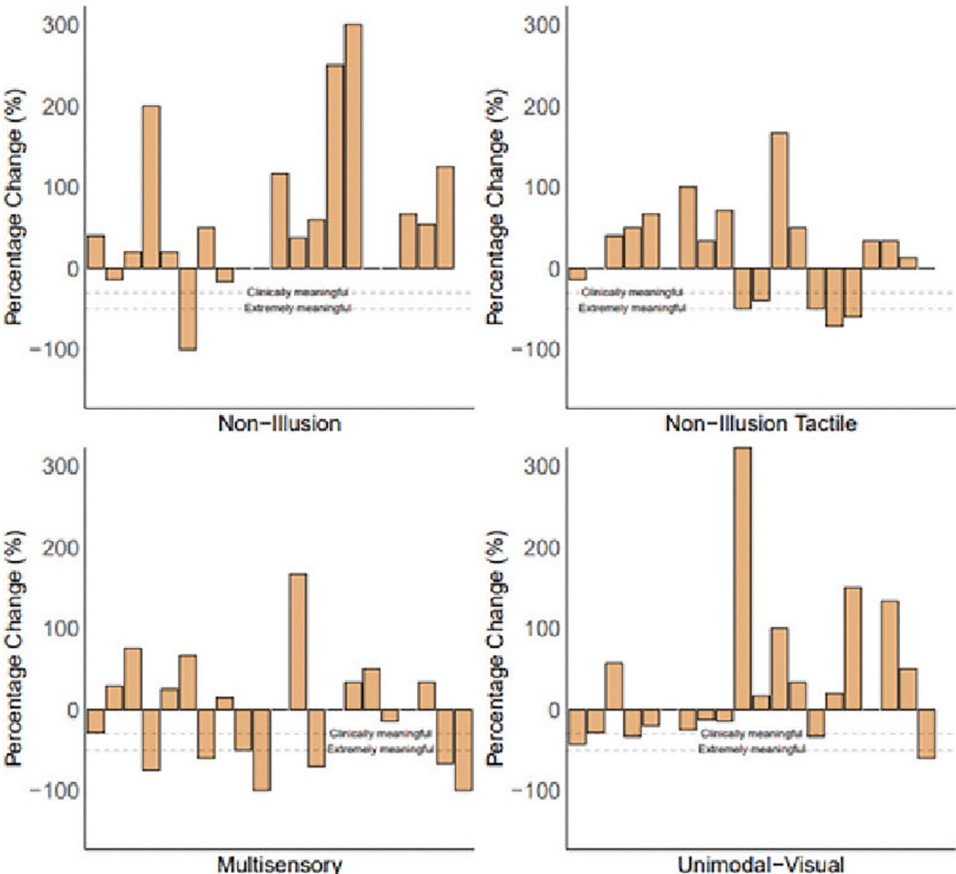

**Fig 6.** (a): Percentage change for pain scores across all conditions per participant. Dashed lines show 30% pain reduction (clinically meaningful) and 50% pain reduction (extremely meaningful).

The somatosensory blurring hypothesis [23] posits that the cortical representation of a painful body part could be blurred and through viewing the body part the representation could become sharpened, which could be the mechanism through which illusory stretching could induce illusory analgesia. Slight reductions in SSEP amplitude have been found before in participants without chronic pain undergoing hand-based resizing illusions [22], however since these participants did not experience chronic pain in their hands it was thought that their cortical representations might not be as blurred as those with chronic hand-based pain such as in the present study. Since no significant difference in SSEP response across resizing conditions were found, it is possible that illusory analgesia could be driven from an alternate mechanism. However, no significant differences in pain levels were found across illusory conditions, meaning SSEP amplitude reductions might not have been expected due to no observed group level illusory analgesia. Since some participants did experience illusory analgesia, exploratory correlations were run between participant's pain percentage change and their SSEP response, to assess whether those experiencing a pain reduction had reduced SSEP amplitudes, however no significant correlations were found.

It is possible that the lack of illusory analgesia seen across participants in the present study was due to the experimental set up. A significant increase in pain was found when comparing pain ratings before and after the non-illusion condition, with some participants commenting that during participation having their hand inside the augmented reality system was painful

for their wrist and shoulder, meaning that they found it difficult to differentiate the pain they were experiencing from the set up itself compared to pain in their manipulated digit. Additionally, participants reported that the need to sit still for 5 minutes per condition block resulted in some additional pain which although reduced through rest breaks between blocks, could have influenced pain ratings taken immediately after a condition.

It is also possible that the vibration elicited during each condition could have reduced participant's pain through a process referred to as vibratory analgesia. Some studies have found vibration to produce up to a 40% reduction in pain intensity [48], whereas others report no significant effects of vibration on pain levels [49]. Due to the therapeutic potential of vibratory analgesia, vibrating gloves have been created as a therapeutic option for people with chronic hand-based pain and have been found to effectively reduce pain levels [50]. Within the present study, it is therefore possible that the 26Hz vibration could have reduced pain levels through vibratory analgesia, however, since vibration was present across all conditions and more participants reported pain reductions in the illusory conditions compared to the non-illusion conditions, it is clear that there were illusory analgesic effects observed beyond that induced by vibratory analgesia. However, this does not mean that the only cause of the observed analgesia is resulting from the illusory manipulations, it is possible that there are other potential cofactors which might influence one's likelihood of experiencing analgesia during illusory resizing, such as possible expectation / placebo effects, which merit further investigation within future research.

Although pain data analyses were conducted at the group level as preregistered, it is important to understand the individual experiences of participants within this sample. Simply because no overall reduction in pain levels were found following either resizing condition (MS or UV), this should not discount the illusory analgesia experiences that several participants reported. Fig 6 shows that a substantial proportion of participants experienced either a clinically or extremally meaningful level of pain reduction following these conditions, highlighting the potential of this therapy for day-to-day treatment. Similarly, however, those experiencing an increase in pain level should not be ignored.It is, therefore, recommended that should illusory resizing be offered as a treatment for chronic hand-based pain that it is not provided within a lab setting such as the current study where significant increases in pain were found due to the experimental set up, but that home based options for delivery of resizing illusions should be prioritised. Future research should assess the potential of mobile phone based illusory resizing, so that it can be delivered from the comfort of one's home. Illusory resizing delivered through a mobile phone would not be able to deliver visuotactile illusions such as the one delivered in the multisensory condition here but could deliver unimodal visual or visual-auditory manipulations. Visual-auditory resizing illusions using a rising pitch tone as non-naturalistic auditory input have been found to increase illusory experiences compared to visual only manipulations [51], therefore both unimodal visual and visual-auditory presentations could be used to deliver meaningful analgesia for a substantial proportion of people living with chronic hand-based pain.

## 5. Conclusions

The present study adds to our understanding of the experiences of illusory resizing within a sample with chronic hand-based pain. The subjective data suggest that people living with chronic pain experience the illusion conditions similarly to those without chronic pain, highlighting the potential of these illusions as a therapeutic treatment avenue. SSEP data however, despite showing a reduction in median amplitude in the multisensory condition in line with the somatosensory blurring / sharpening hypothesis, did not show overall significant

differences between conditions possibly due to the lack of illusory analgesia experienced for some within the sample. This lack of group level amplitude reductions during illusory conditions, however, does not mean support was found for the magnification hypothesis, since no significant increases in median SSEP amplitudes during illusory conditions were observed either. Pain data highlighted the individual nature of chronic pain, with group analyses showing no significant effects, but participant data showing strong experiences of both pain reduction and pain increases. This individuality was not underpinned by type of chronic pain, with both chronic primary and secondary pain condition subgroups showing no significant differences in pain percentage change. These nuances of chronic pain experiences must be considered when designing therapies for pain alleviation, to ensure people understand how varied analgesic effects from resizing illusions can be.

## Supporting information

**S1 Fig. Averaged pilot data showing peak frequency at 26Hz, centred between electrodes F1 and FC1.**
(PDF)

**S2 Fig. Averaged Illusion score for each condition.**
(PDF)

**S3 Fig. Combined disownership score index across conditions (NI: Non-Illusion; NIT: Non-Illusion Tactile; MS: Multisensory; UV: Unimodal Visual).**
(PDF)

**S4 Fig. Combined control scores across conditions (NI: Non-Illusion; NIT: Non-Illusion Tactile; MS: Multisensory; UV: Unimodal Visual).**
(PDF)

**S5 Fig. Correlation between amplitude and subjective Illusory score for each condition.**
(PDF)

**S6 Fig. Correlation between amplitude and pain percentage change for each condition.**
(PDF)

**S7 Fig. Correlation between pain percentage change and subjective illusory score for each condition.**
(PDF)

**S8 Fig. Pre and post pain scores across conditions for participants with chronic primary pain.**
(PDF)

**S9 Fig. Pre and post pain scores across conditions for participants with chronic secondary pain.**
(PDF)

**S10 Fig. Percentage change for pain scores across all conditions per participant.**
(PDF)

## Acknowledgments

The authors would like to thank B.P.A Quinn for production of the schematic used in Fig 1. We would also like to thank all the participants who gave their time to take part in this study.

## Author Contributions

**Conceptualization:** Kirralise J. Hansford, Daniel H. Baker, Catherine E. J. Preston.

**Data curation:** Kirralise J. Hansford, Daniel H. Baker.

**Formal analysis:** Kirralise J. Hansford.

**Funding acquisition:** Catherine E. J. Preston.

**Investigation:** Kirralise J. Hansford.

**Methodology:** Kirralise J. Hansford, Daniel H. Baker, Catherine E. J. Preston.

**Project administration:** Kirralise J. Hansford, Catherine E. J. Preston.

**Resources:** Daniel H. Baker, Kirsten J. McKenzie, Catherine E. J. Preston.

**Software:** Kirralise J. Hansford, Daniel H. Baker.

**Supervision:** Daniel H. Baker, Kirsten J. McKenzie, Catherine E. J. Preston.

**Validation:** Kirralise J. Hansford, Daniel H. Baker.

**Visualization:** Kirralise J. Hansford.

**Writing – original draft:** Kirralise J. Hansford.

**Writing – review & editing:** Daniel H. Baker, Kirsten J. McKenzie, Catherine E. J. Preston.

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
