## [Decision Letter · Decision Letter 0]

28 Oct 2024

PONE-D-24-37256Illusory finger stretching and somatosensory responses in participants with chronic hand-based painPLOS ONE

Dear Dr. Hansford,

Thank you for submitting your manuscript to PLOS ONE. The reviewers are positive about your paper but have a number of recommendations that need to be addressed in a revision. Therefore, we invite you to submit a revised version of the manuscript that addresses the points raised during the review process.

We look forward to receiving your revised manuscript.

Kind regards,

Jane Elizabeth Aspell, PhD

Academic Editor

PLOS ONE

Journal Requirements:

The authors would like to thank the Pain Relief Foundation for funding this research, and B.P.A Quinn for production of the schematic used in Figure 1. We would also like to thank all the participants who gave their time to take part in this study. 

 A Pain Relief Foundation (https://painrelieffoundation.org.uk/) John Miles PhD Studentship grant funded the work for this manuscript. The grant was awarded to C.E.J.P. The funders did not play any role in study design, data collection or analysis, or the preparation of the manuscript. 

Reviewers' comments:

Reviewer's Responses to Questions

**Comments to the Author**

1. Is the manuscript technically sound, and do the data support the conclusions?

Reviewer #1: Yes

Reviewer #2: Yes

2. Has the statistical analysis been performed appropriately and rigorously? 

Reviewer #1: Yes

Reviewer #2: Yes

3. Have the authors made all data underlying the findings in their manuscript fully available?

Reviewer #1: Yes

Reviewer #2: Yes

4. Is the manuscript presented in an intelligible fashion and written in standard English?

Reviewer #1: Yes

Reviewer #2: Yes

5. Review Comments to the Author

Reviewer #1: In this manuscript, the authors conducted a study to evaluate the potential analgesic effect of visuotactile resizing finger illusions in a population suffering from chronic pain. This study not only builds upon a previous study conducted in populations without chronic pain but also presents a significant potential impact for its clinical applications. The authors also sought to explore the possible correlation between Steady-State Evoked Potentials (SSEP) magnitudes and illusory resizing, aiming to assess the potential link between SSEPs and the analgesic effect of resizing illusions. While brain activity (SSEP) showed no significant changes during the illusions, some participants experienced meaningful pain relief, especially in multisensory condition. The study suggests that resizing illusions could offer targeted pain reduction for some individuals, though the exact neural mechanisms remain unclear. The analyses were carried out using established methods and questionnaires to measure illusory strength, SSEP magnitude, and pain scale before and after interventions for the affected joint/s.

The study was pre-registered, with explicitly stated hypotheses and a well-articulated rationale. Both the methods and results are presented in a straightforward manner. However, the reviewers believe that addressing a few key points could enhance the paper's impact and improve its overall coherence.

• There is conflicting information between the description of chronic pain diagnosis (primary, secondary, and non-diagnosed) in lines 154-155 and the inclusion criteria in lines 169-170, which state that no formal diagnosis was required. Was any type of diagnosis conducted before the study? If no formal diagnosis was performed and participants with non-diagnosed chronic pain were included based on self-reported joint pain lasting three or more months, this should be stated more explicitly to clarify the characteristics of the study population.

• If no formal diagnosis of chronic pain was required for participation in the study, this would imply a significant derivation from the inclusion criteria stated in the pre-registered document at https://osf.io/9anjc. Including a section indicating the possible derivations from the pre-registered document, including the decision not to acquire further datasets after excluding a participant, would be desirable to reinforce the power and relevance of the results presented in this manuscript.

• The reviewers understand that a subset of participants was excluded because they had been previously diagnosed with Complex Regional Pain Syndrome (lines 179-180). It looks some participants were previously given a Diagnosis? Please clarify

• Line 279-281: The reviewers find a bit unclear why the first two questions are grouped under "illusory experience." Grouping the questions into Ownership (e.g., "It felt like my finger was really stretching" / "It felt like the finger I saw was part of my body") and Disownership (e.g., "It felt like the finger I saw no longer belonged to me" / "It felt like the finger I saw was no longer part of my body") might provide a clearer distinction between the positive experience of ownership and the sense of losing ownership. This would make it easier for readers to grasp the different sensory perceptions being measured.

- There is a possible error in the reference index for the study of Gilpin et al. in line 106, where the reference index is set at 24 while that same index corresponds to an uncited article by Haggard et al. in the reference list.

• Data analysis: The authors performed Wilcoxon tests to compare pre- and post-pain levels across conditions, which is suitable for non-parametric data but primarily addresses group-level effects. However, given that some individuals experienced clinically meaningful pain reductions (greater than 30% or 50%), this method may overlook important individual variability in responses. A generalized linear mixed model (GLMM) would allow the authors to account for both group and individual-level variability, offering a more comprehensive analysis. By using GLMM, they could better capture the differences in pain reduction across participants, potentially revealing significant effects that the Wilcoxon test, focused on overall group trends, might not detect. This would provide a clearer picture of how the illusory conditions impacted pain on an individual basis, especially for those who experienced substantial pain relief.

• While Illusory Resizing showed strong analgesic effects in some individuals, the lack of a consistent effect across all participants suggests that illusory analgesia may be present but cannot be definitively attributed to the manipulation alone. Other potential cofactors cannot be ruled out, especially given the absence of statistically significant group-level differences, contrary to the statement in lines 526-527. Please argument.

• In the conclusion (around line 552), the SSEP data showed a reduction in median amplitude in the multisensory condition, aligning with the somatosensory blurring/sharpening hypothesis. However, no significant differences were found between conditions, possibly due to some participants not experiencing illusory analgesia. If the somatosensory blurring/sharpening hypothesis is not fully supported, this raises the question: Is the theory suggesting that magnifying the hand to increase its cortical representation, and thereby reduce pain, a more fitting explanation? Or are there other possible explanations? The authors should clarify this point.

Other minor points:

• The sentence in lines 63-65 could be re-arranged or split into two sentences to increase readability.

• The presentation of the research hypotheses in the introduction would be clearer if they were explicitly grouped into three categories: into Illusory Experience, SSEP, and Pain Reduction to simplify their comprehension.

• The statement about the importance of applying Illusory Resizing in clinical settings (lines 534-536) would have a greater impact if it were separated into its own sentence, apart from the argument about its lab use (lines 536-537).

Reviewer #2: The study by Hansford and colleagues aimed to test the effects of (multi)sensory illusions on reducing chronic hand pain. Four different experimental conditions were designed, where proprioceptive, visual, and haptic feedback were either combined or presented individually in an augmented virtual reality setting. Participants were asked to assess their pain and sense of hand disownership before and after each condition. Additionally, steady-state evoked potentials (SSEPs) were recorded to explore potential changes related to the illusory analgesia.

The results showed a stronger resizing illusion for the multisensory integration condition compared to the two control non-illusion conditions, though no significant effects were found on SSEPs. Although some participants experienced pain reduction following the visuo-proprioceptive illusion, this effect was not significant at the group level.

I found that the study appropriately explored its aims and tested the hypotheses presented by the authors. I also appreciate the open science approach. However, some minor points should be addressed before I can recommend it for publication:

1 Previous studies (e.g., Tidoni et al., 2015) found that the physical parameters of tendon vibration could be individually adjusted to induce optimal illusions. This has also been observed in clinical samples (e.g., spinal cord injury, Fusco et al., 2016). Given this, how and why did the authors select 25 Hz and 0.18 N to induce the digit resizing illusion?

2 Since proprioceptive illusions can reflect different sensitivity or susceptibility across participants, did the authors introduce the type of illusion beforehand to establish any kind of prior expectation?

3 The authors reported that SSEPs were averaged across the activity recorded during the entire trial window. However, since the onset of the illusion may vary from trial to trial and might not correspond to time 0 (i.e., the beginning of the trial), could the null effect observed be due to varying patterns that do not consistently reflect the illusion?

4 Related to the previous point, when did the experimenter begin pulling the finger, and how was this aligned with the individual onset of the illusion? Did participants verbally indicate when they started feeling the illusion?

5 Why did the experimenter touch the target finger without pulling it? Please clarify this aspect.

6. PLOS authors have the option to publish the peer review history of their article (what does this mean?). If published, this will include your full peer review and any attached files.

Reviewer #1: No

Reviewer #2: No

---

## [Author Response · Author response to Decision Letter 0]

4 Nov 2024

Response to reviewers has been uploaded as a word document file.

---

## [Decision Letter · Decision Letter 1]

3 Jan 2025

Illusory finger stretching and somatosensory responses in participants with chronic hand-based pain

PONE-D-24-37256R1

Dear Dr. Hansford,

We’re pleased to inform you that your manuscript has been judged scientifically suitable for publication and will be formally accepted for publication once it meets all outstanding technical requirements.

Kind regards,

Prof. Jane Elizabeth Aspell, PhD

Academic Editor

PLOS ONE

Reviewers' comments:

Reviewer's Responses to Questions

**Comments to the Author**

1. If the authors have adequately addressed your comments raised in a previous round of review and you feel that this manuscript is now acceptable for publication, you may indicate that here to bypass the “Comments to the Author” section, enter your conflict of interest statement in the “Confidential to Editor” section, and submit your "Accept" recommendation.

Reviewer #1: All comments have been addressed

Reviewer #2: All comments have been addressed

2. Is the manuscript technically sound, and do the data support the conclusions?

Reviewer #1: Yes

Reviewer #2: Yes

3. Has the statistical analysis been performed appropriately and rigorously? 

Reviewer #1: Yes

Reviewer #2: Yes

4. Have the authors made all data underlying the findings in their manuscript fully available?

Reviewer #1: Yes

Reviewer #2: Yes

5. Is the manuscript presented in an intelligible fashion and written in standard English?

Reviewer #1: Yes

Reviewer #2: Yes

6. Review Comments to the Author

Reviewer #1: Thank you for addressing the points carefully and meticulously. The reviewer acknowledges the quality of the paper, which in its present form is now ready for publication in PLOS ONE.

Reviewer #2: I thank the Authors for having addressed all the points. I think the manuscript is now suitable for the pubblication . Best Wishes

7. PLOS authors have the option to publish the peer review history of their article (what does this mean?). If published, this will include your full peer review and any attached files.

Reviewer #1: No

Reviewer #2: **Yes: **Gabriele Fusco

---

## [Editor Report · Acceptance letter]

14 Jan 2025

PONE-D-24-37256R1 

PLOS ONE

Dear Dr. Hansford, 

I'm pleased to inform you that your manuscript has been deemed suitable for publication in PLOS ONE. Congratulations! Your manuscript is now being handed over to our production team.

Kind regards, 

on behalf of

Prof. Jane Elizabeth Aspell 

Academic Editor

PLOS ONE